# Examination of Primary and Secondary Metabolites Associated with a Plant-Based Diet and Their Impact on Human Health

**DOI:** 10.3390/foods13071020

**Published:** 2024-03-27

**Authors:** Miray Simsek, Kristin Whitney

**Affiliations:** 1Department of Biochemistry, Purdue University, West Lafayette, IN 47907, USA; 2Department of Food Science and Whistler Center for Carbohydrate Research, Purdue University, West Lafayette, IN 47907, USA

**Keywords:** plant, bioactive components, chronic diseases, human health

## Abstract

The consumption of plant-based diets has become a burgeoning trend, and they are increasingly consumed globally owing to their substantial energy intensity and dietetic advantages. Plants possess numerous bioactive components that have been recognized to exhibit manifold health-promoting assets. Comprehension of the synthesis of these primary and secondary metabolites by plants and their method of action against several chronic illnesses is a significant requirement for understanding their benefits to human health and disease prevention. Furthermore, the association of biologically active complexes with plants, humans, disease, medicine, and the underlying mechanisms is unexplored. Therefore, this review portrays various bioactive components derived from plant sources associated with health-promoting traits and their action mechanisms. This review paper predominantly assembles proposed plant-derived bioactive compounds, postulating valuable evidence aimed at perceiving forthcoming approaches, including the selection of potent bioactive components for formulating functional diets that are effective against several human disorders. This meticulous evidence could perhaps provide the basis for the advanced preemptive and therapeutic potential promoting human health. Hence, delivery opens possibilities for purchasers to approach the lucrative practice of plants as a remedy, produce novel products, and access new marketplaces.

## 1. Introduction

Nutrition is a vital factor that moderates the equilibrium between healthy and diseased states. Food offers various nutrients that are indispensable for living, in addition to holding potential bioactive attributes that are involved in human health and disease defense mechanisms [1]. With the increased consumer awareness regarding the nutritious and therapeutic potentials of fruits and vegetables, consumer emphasis is more on improving the chemical constitution of plants [2]. In addition to numerous macro- and micronutrient components that are essential for a healthy human diet, plants are a rich source of bioactive compounds. A wide range of metabolites (primary and secondary) produced by plants positively impact human health and sustenance. The compounds directly involved in the plant’s functional and biological processes are termed primary metabolites. The primary metabolites vital for human sustenance are carbohydrates, lipids, proteins, and vitamins [3]. Compounds produced by plants in response to various environmental stresses are termed secondary metabolites. Secondary metabolites do not participate in normal plant processes [4]. Secondary metabolites are synthesized by a controlled system, the condition when plants do not experience any stress is the equilibrium, and the redox state appears as an indicator permitting plants to execute normally. However, if the plant experiences a challenge, the redox state changes, indicating that the plant needs to adapt to account for stress by stimulating the production of metabolites to acclimate to the stress conditions. In extreme stress conditions, plant and redox states change to a higher degree, perhaps initiating apoptosis feedback in plants instigating aging and, ultimately, death [5].

In a nutshell, plant-derived metabolites perform indispensable roles in the plant defense system against abiotic and biotic environmental stresses. Abiotic stress includes temperature, light, wounding, chemical derivatives, and so on, whereas biotic stress is associated with microbes, insects, and animals [6]. The biosynthesis of these compounds is via several biochemical pathways, controlled via a signal transduction path involving indicators, transcription factors, expression of various genes linked with enzymes synchronizing the respective pathways, several genes attenuating expression of another gene, and cross-talks amid pathways on biomolecular stages [7]. While plants synthesize these complexes for their own use and benefit, humans have been exploiting these complexes, specifically the health-promoting secondary metabolites, over the centuries [8]. Modern plant-based diets comprise vegetables, fruits, grains, and legumes, which remain well endowed with several beneficial compounds: bioactive constituents (carotenoids, phytochemicals, dietary fiber, and vitamins) and micronutrients (calcium, magnesium, and potassium) [9]. Each of the compounds holds diverse attributes concerning human sustenance and health (Table 1).

Plant-derived antioxidants, mainly polyphenols, for example, flavonoids, lignans, stilbenes, and phenolic acids, possess a comprehensive array of biological traits, antiaging, anticancer, anti-infectious, anti-inflammatory, antiviral, etc. [36,37]. There are also several micronutrients, including vitamins (fat-soluble and water-soluble), minerals, and trace elements that are required for human health. These compounds have a wide array of biological functions within the human body [32]. For those consuming plant-based diets, it is important to ensure a balanced diet in order to obtain the correct amounts of these critical nutrients. However, even those on omnivorous diets can present some deficiencies if a balanced diet is lacking [38].

Similarly, food-derived bioactive peptides develop expanded prominence owing to various biologically active assets, which are stated to be reliant on respective amino acid patterns [23,24,25]. The bioactive peptides and definite amino acid patterns display expedient biological potency or exert an affirmative impact on the functioning of the human body, irrespective of their nutritional status [25]. It is well documented that plant proteins comprise bioactive peptides as an inactive amino acid pattern once existing in the native protein. Nonetheless, bioactive peptides could actively discharge through different processes such as fermentation, in vitro or digestive tract, enzymatic proteolysis, etc. [39]. Additionally, there are biologically important fatty acids that are well documented and continue to be studied for their health-promoting effects. For example, ω-3-fatty acids are a common supplement for which there are several plant sources [40,41].

Lately, wide investigations have focused on the application of different types of diet patterns. Likewise, many of the reviews focus only on investigating the impacts of plant compounds treating various chronic diseases with inadequate diverse pathways. Limited researchers evaluate their effects on human health at biological and molecular levels. As far as we know, fragmented information is available concerning the portrayal of respective plant biochemical complexes assisting in human growth and development with diverse pathways. Moreover, the association between plants, their compounds, and human diseases is yet not well outlined. Comprehending how the molecular targets of plant-derived compounds along with their action mechanism on human health and physiology is necessitated. Additionally, extensive scientific publications on scientific platforms and public vogue portrayed by Google accentuate escalated captivation in plant-based regimens. Thus, such escalating vigilance demands a protracted scientific appraisal of how plant-based diets influence human health, particularly with reverence to possible molecular action mechanisms.

Therefore, in this review paper, our objective is to synopsize the portrayal of the association of several plant compounds with protection in relation to the cause and the preclusion of several chronic disorders. The scope of this review includes (i) the beneficial role of biochemical active composites in plants and humans, (ii) plant-based compounds association with the incidence of chronic diseases, (iii) action mechanism accountable for the consumption of plant-derived compounds with major health outcomes, (iv) exploration of why and how plant-based compounds are associated with a low prevalence of chronic disorders: prospective and pragmatic associations. Although research emphasizes a particular component of plants responsible for inhibiting respective disorders, the present article states that this inhibition is not established alone with a single compound. Somewhere, the assistance of another plant compound also emerges. Furthermore, a description of the hypothesis of how stress, plants, and humans are correlated to each other, along with proposing a holistic framework associated with food, chronic diseases, and medicine, is also established. This review article postulates a synopsis of the repercussions of the consumption of a plant-origin diet, concisely focusing on nutritionary assets and their correlation in improving several health disorders. Overall, the goal and scope of this review is to provide a holistic overview of the contribution of plant-based diets to the consumption and health benefits of plant metabolites and phytochemicals.

## 2. Associations of Plant-Based Diets and the Associated Beneficial Compounds with the Prevalence of Chronic Diseases

The term “plant-based diet” has been in use for over two decades and has been the topic of discussion and research among the public and professionals. However, there is no official definition for a plant-based diet, and this term may refer to diets with a low consumption of meat, various types of vegetarian diets, or vegan (100% plant) diets [42]. Many people consider plant-based diets to be healthier than omnivorous diets containing a large amount of meat products. Some research has shown an association between plant-based diets and all causes of mortality and a moderately lower risk of death [43,44,45]. The increasing popularity of plant-based diets and the possible health benefits suggest that additional work related to specific compounds prevalent in plant-based diets will be important to identify the key compounds confirming such health benefits.

Plant-based foods have displayed several benefits to mankind, including bioactive peptides and phenolic compounds, in addition to enhanced dietary health index. Hence, what is the exceptional association between plants and human health? Principally, when plants experience stress (e.g., oxidative stress), they synthesize certain chemical complexes that aid in protection and adaptation against those stressors. For example, plants under salt stress can produce metabolites, such as bioactive peptides, to maintain low intracellular osmotic potential [46]. Another example is the production of phenolic compounds in plant tissue as part of an adaptive response to stressors. Stress conditions that may illicit the production of phenolic compounds in plants include low temperature, pathogen infection, nutrient deficiency, and herbivores [47]. However, in humans, when their cells/tissues experience some stress, such as oxidative stress, it generates inflammation, thereby instigating potential chronic diseases associated with it [48]. Thus, when humans ingest these chemical compounds, they can ameliorate inflammation and the diseases corresponding to it (Figure 1).

### 2.1. Plant Protein

Plant proteins mainly serve the purpose of offering suitable nourishment and are largely advocated in diverse disease conditions, including diabetes, heart diseases, obesity, cancer, etc. The inaccessibility of nutritive, satisfactory, and protected quality food subsequently leads to a state of undernourishment and various illnesses. Protein deficiency in plant-based diets is a major concern related to undernourishment [49]. Even though essential nutrients are maintained, malnourishment occurs if the delivery and distribution of proteins or other nutrients are not maintained. Growing knowledge concerning the importance of proteins in the diet encourages researchers/nutritionists to investigate sustainable sources of protein. Concerning this, plant-derived proteins are progressively exploited over other protein sources in human diets [50].

An important factor to consider regarding plant proteins is their digestibility compared to animal sources of protein. There has been a consensus that plant proteins are generally less digestible than animal proteins [51]. Many plant sources of protein contain antinutritional factors such as trypsin inhibitors, β-mannans, and glucosinolates, which reduce amino acid digestibility [52]. Some examples of plants that produce these antinutritional factors are soybean, quinoa, almonds, pomegranate, wheat, green tea, and cabbage [53]. One in vitro study of the digestibility of plant proteins compared to milk protein showed that the digestion of legume protein isolates, except for soybean, was similar to that found for milk proteins [51].

One study comparing an exclusively plant-based diet and a protein-matched mixed diet showed that there was no difference in supporting muscle mass and mass accrual [54]. There is some evidence that the risk of exacerbating type 2 diabetes may be reduced by intake of plant-based proteins. In a study of newly diagnosed type 2 diabetes patients without glucose lowering treatment, the likeliness of remittance from diabetes was increased for those who increased their intake of plant-based protein. However, it was noted that these outcomes may also be partially related to other components in the diets with increased plant protein, such as dietary fiber and micronutrients [55].

Additionally, Tong et al. [56] found that dietary proteins had an effect on cholesterol metabolism, which was dependent on susceptible gut microbiota. Upon metabolomic analysis, it was found that cholesterol-lowering effect of the pea protein was related to changes in the gut microbiome and improvement of amino acid metabolism [56]. Researchers also found that hempseed proteins carry out antioxidant and anti-inflammatory activities in HepG2 cells, which is related to possible hypercholesteremic effects. Augmentation of the low-density lipoprotein receptor (LDLR) localized on the cellular membranes resulted in an improved ability of HepG2 cells to uptake extracellular low-density lipoprotein (LDL). A specific peptide from hemp seed was shown to have the potential for cardiovascular disease (CVD) prevention [57].

A systemic review and meta-analysis from the National Osteoporosis Foundation did not find any significant evidence that there was any difference between the consumption of soy protein and animal protein with regards to bone health. However, it was determined that additional investigation is needed due to confounding factors presented in several of the studies included in the meta-analysis [58]. Overall, various types of studies show differing effects and modes of action for the impact of plant protein on health. However, this area requires additional study, especially for long-term human clinical studies.

### 2.2. Phytochemicals—Phenolic Compounds

Phytochemicals refer to the wide array of secondary metabolites produced by plants, many of which are phenolic compounds (Table 1). There is evidence that phytochemicals from plants have health promoting effects [59]. Many plant foods such as soybean, cranberry, eggplant, grape, hemp grain, quinoa, and sorghum are rich in different phenolic compounds. Some of the main modes of action for several classes of dietary phenolic compounds are regulation of reactive oxygen species (ROS), antioxidant activity, and anti-inflammatory activity [59]. Naringin is a phytochemical with a typical flavonoid structure having three rings connected by a three-carbon chain and two rhamnose units attached at C7, and naringenin has the same structure without the rhamnose substituent. The structure of these flavanone compounds results in their antioxidant activity. Naringin has low bioavailability and is converted to naringenin in the gut. However, naringin is a safe and nontoxic compound found in citrus fruits, and it has been reported to have a therapeutic concentration of approximately 300 mg (two times per day). Studies have shown that these compounds may have beneficial impacts on inflammation, autophagy, apoptosis, cell proliferation, and angiogenesis [10].

Another study investigated the impact of cocoa flavanols on cerebral cortical oxygenation and cognition. In a randomized, double-blind within-subject acute study in healthy young adults, Gratton et al. [11] found that the intake of flavanol led to improved cognitive function. The study found that high flavanol intake was linked to a greater oxygenation response. Overall, this study was able to show the benefit of flavanols on cognitive function in healthy adults [11].

Diets high in phenolic compounds from dietary sources such as olives, olive oil, and red wine are purported to be a factor in the prevention of CVD. A study following participants over a median of 12.2 years found a higher risk of CVD for those with a low (432.1 mg/day phenolics and 190.7 mg/day flavonoid) intake of dietary phenolic compounds. Additionally, there was a likely reduction in CVD incidence for those having a Mediterranean diet with moderate to high intake of phenolic compounds. The evidence also indicated that a five percent reduction in the risk of CVD incidence could be achieved by increasing the phenolic dosage by 10 mg/day [13].

Lignin has been proposed to have health benefits; it is one of the most abundant phenolic compounds found in plants, and it has antioxidant properties due to its action in countering lipid peroxidation [16,18]. There is some evidence to suggest that sulfated lignin may be a viable candidate to prevent or treat type 2 diabetes mellitus. One type of sulfated lignin presented an inhibition capacity of α-amylase (half-maximal inhibitory concentration (IC_50_), 32.35 μg/mL), which was similar to acarbose (IC_50_, 27.33 μg/mL, a common medication for type 2 diabetes management). Additionally, the sulfated lignin was found to have favorable effects on the population of beneficial gut bacteria [16].

Inflammatory bowel disease (IBD) is a multifactorial disease that encompasses two major conditions: ulcerative colitis and Crohn’s disease. These conditions result in inflammation in part or all of the gastrointestinal tract. Currently, there are several treatments to mitigate the symptoms of these conditions, but a preventative or curative agent has not been developed. The synergistic effects of lignin and *Lactobacillus plantarum* against dextran sodium sulphate (DSS)-induced IBD mouse was investigated. The mice treated with lignin and *L. plantarum* had a markedly reduced disease index compared to the DSS induce mice [18]. 

The treatment with lignin and *L. plantarum* also impacted the levels of cytokines, which are generated by lactic acid bacteria and play crucial roles in immunological modulation. Cytokine levels increased in DSS treated mice, but all cytokines in the study decreased for the lignin, *L. plantarum,* and ligin+ *L. plantarum* treatments [18]. Overall, there are some indications that different types of lignin may be useful for amelioration of conditions such as diabetes or irritable bowel syndrome (IBS). Ultimately, there are varied degrees of evidence for the importance of phytochemicals in the human diet. Plants are a valuable source of phytochemicals and it will be important to conduct longer-term human studies on their impact on various aspects of human health.

### 2.3. Micronutrients

While large amounts of macronutrients such as protein, fat, and carbohydrates are needed by the body, micronutrients are those that are only needed in very small amounts (Table 1). The amount of each micronutrient needed for a healthy diet depends on metabolic condition and stage of life. Even though only small amounts of these nutrients are required, they are still essential for a healthy diet, and deficiencies may cause deleterious effects [26,32]. Micronutrients can be classified into four main groups: fat-soluble vitamins (A, D, E, and K), water-soluble vitamins (B1, B2, B3, B5, B6, B7, B9, B12, and C), macro-minerals (Ca, Na, Mg, K, and P), and trace elements (S, Fe, Cl, Co, Cu, Zn, Mo, I, and Se) [32]. Many of the fat-soluble vitamins are found primarily in animal products, so those eating a plant-based or vegan diet need to consume enough of these nutrients. For example, preformed vitamin A is mainly found in animal products such as liver, egg yolk, and milk. However, pro-vitamin A can be found in green leafy vegetables, carrots, mangoes, and sweet potatoes [32].

An important group of water-soluble vitamins for those on plant-based or vegan diets are the B vitamins. The B vitamins have a variety of biological effects such as neurological functions, conversion of food into energy, production of cell’s genetic material, red blood cell formation, and regulate metabolism of protein, fats, and carbohydrates [29,30]. Those consuming a completely vegan diet will generally require the intake of vitamin B supplements as vegan diets contain very low amounts of B vitamins and almost no vitamin B12 [30]. There is also evidence of a connection between B vitamins and Parkinson’s disease. Parkinson’s disease symptoms may worsen for those having low B12 levels [60]. Another study showed that while long-term elevated intake of vitamin B6 or folate does not reduce the risk of Parkinson’s disease, there may be a possible protective effect from vitamin B12 [61]. Supplementation with B vitamins can prevent the adverse health effects of dietary deficiency [30]. Overall, it is essential that a well-balanced and healthy diet contains the necessary amounts of all macro- and micronutrients. 

## 3. Mechanisms Associating the Consumption of Plant-Based Compounds with Health Outcomes

There has been a significant amount of research showing the benefits of consuming a diet rich in plant foods and their association with lowered risk of CVD and all-cause mortality. However, there have been some conflicting reports regarding how plant-based diets compare to omnivorous diets regarding disease and mortality risks [43]. One study on the general US adult population indicated that diets with a higher proportion of plant foods and lower proportions of animal foods were linked to a lower incidence of CVD, CVD mortality, and all-cause mortality. Those who are in the highest quintiles of plant-based diet index (PDI), healthy plant-based diet index (hPDI), and provegetarian diet index were observed to consume significantly more fruit and vegetable servings per day while reducing meat servings. The increase in fruit and vegetable consumption along with the reduction in meat was linked to increased intake of dietary fiber and micronutrients and reduced saturated fat and cholesterol [62]. Along with the overall evidence supporting the benefits of plant-based diets on health, work has been carried out to determine the mechanisms behind the benefits of plant-based diets and their associated beneficial compounds.

There are complex interdependencies between components of different types of diets and their physiological functions. Thus, it is improbable that the benefits of vegan diets simply come down to lower calories. The beneficial compounds found in plants, which would be consumed in higher amounts for vegan or plant-based diets, probably act through multiple pathways. These pathways include improved glycemic control, lower inflammatory activity, and altered neurotransmitter metabolism [63]. While the exact mechanisms of plant rich diets such as the Mediterranean diet are not known, research has shown five main mechanisms by which the Mediterranean diet results in positive health outcomes. These mechanisms include (1) lipid-lowering effects, (2) protection against oxidative stress (inflammation and platelet aggregation), (3) modification of hormones and growth factors related to cancer, (4) inhibition of nutrient sensing pathways (specific amino acid restriction), and (5) gut-microbiota-mediated production of metabolites related to metabolic health. A specific example of one of these mechanisms is the health-promoting and prolongevity effects from moderate energy restriction which is attained due to specific restriction of sulfur and branch chain amino acids, and saturated fatty acids when consuming fiber-rich/energy-poor foods common in plant-based diets [64].

Nevertheless, conventional or modern medications are well recognized as more efficacious in reversing the diseased condition to the predisease stage by the administration of various therapeutic approaches, but certainly not to the normal health state [65]. Generally, conventional medications are inapplicable for prevention. The holistic process of altering dietary intake discloses several potentialities of discovering plant-based complexes as nutraceuticals through an innovative notion (Figure 2).

### 3.1. Mechanisms of Plant-Derived Metabolites in the Intestinal System

The modes of action of plant-derived metabolites, such as phenolic compounds and bioactive peptides, have been studied for their impacts on the intestinal system and gut health. It is well known that gut microbiota play a crucial role in human health and disease. Phenolic compounds such as flavonoids impact the gut microbiome, resulting in health impacts. The mechanism of how flavonoids affect gut health has been studied via fecal fermentation. During in vitro fecal fermentation, hesperitin-7-O-glucoside, prunin, and isoquercitrin reduced gas production as compared to the control, which produced a large amount of CO_2_, H_2_, and small amounts of H_2_S, CH_4_, and NH_3_ [66]. These and other flavonoids are common in citrus fruits [67].

Pan et al. [66] also found that the main mode of action of citrus flavonoids was their impact on short-chain fatty acid (SCFA) production and the composition of fecal microbiota after fermentation. Specifically, the addition of the citrus flavonoids lowered the production of SCFA to 15 mmol/L or lower, as compared to the control at 23 mmol/L. There was also a significant (*p* < 0.05) reduction in acetic and propionic acid production with the addition of the prunin and hesperitin-7-O-glucoside. Ultimately, the changes in SCFA contents were related to shifts in gut microbial populations, and the decrease in SCFA was related to the broad-spectrum inhibitory effect of the flavonoids on the gut bacteria [66]. While there is some conflicting evidence of the effect of flavonoids on gut bacteria, it is important to understand how these compounds impact different microbial species and that each individual person has a unique gut microbiome. Thus, utilization of compounds such as flavonoids to alter gut health may need to be conducted on an individual basis rather than on a population-wide basis.

Another recent study found that dietary phenolic acids play a role in alleviating intestinal inflammation. It has been proposed that the presence of phenolic acids alters the composition of the gut microbiota and regulates macrophage activation. Han et al. [68] conducted a study on the mode of action by which phenolic acids impact gut microbiota and macrophages related to gut inflammation and IBS. The severity of dextran sodium sulfate (DSS)-induced colitis in mice was reduced with oral administration of chlorogenic acid, caffeic acid, ferulic acid, and ellagic acid. Administration of phenolic acids also significantly reduced secretion of the inflammatory markers tumor necrosis factor alpha (TNF-α), interleukin 1 beta (IL-1β), and interleukin-6 (IL-6). The different phenolic acids showed variation in the mechanism in which they protect against colitis. Chlorogenic acid was shown to have a macrophage-dependent mechanism against colitis, while ferulic acid reduced colitis at least partially through a neutrophil-dependent mechanism. Additional phenolic acids showed other modes of action against colitis, and, thus, it is important to understand how each phenolic acid interacts with the gut microbiome and what its impact is on macrophages [68].

Lignin is the second most prevalent natural aromatic polymer found in natural ecosystems after cellulose and is a prebiotic with antioxidant potential. Kaliyamoorthy et al. [18] studied the mechanistic synergism between lignin and probiotics (*Lactobacillus plantarum*) in a DSS colitis model. Upon treatment with DDS, there were several negative effects, which were reversed in the lignin- and *L. plantarum*-treated groups. The results of the study showed that supplementation was able to inhibit the increase in DSS-induced cytokine levels. Additionally, the expression of E-cad was restored while STAT3 was suppressed as a result of the lignin and *L. plantarum* supplementation according to the gene and protein expression study. Overall, it was determined that lignin and *L. plantarum* have a synergistic role in the protection against IBS by changing the inflammatory cytokines and key gene expression for DSS-induced colitis. The results of the study indicated that the mechanism behind the synergistic action of lignan and *L. plantarum* involves recovery of intestinal damage and suppression of colitis recurrence. These mechanisms are an effect of reduced inflammation (inhibition of inflammatory cytokines) [18].

On the whole, micronutrients and phytochemicals are essential for a healthy gut system and, thus, for overall human health. The gut microbiome is a complex system that affects not only digestion but the inflammatory responses, dysregulation of bodily function, and risks for chronic disease. Recent studies have shown the mechanism of action that bioactive compounds have on the intestinal system and gut health through their complex interaction with gut microbiota and the biochemical processes in the intestinal system [18,66,68]. Consuming plant foods rich in phytochemicals, such as phenolic compounds, in addition to dietary fiber, is important for sustaining a healthy gut microbiome and maintaining homeostasis.

### 3.2. Mechanisms of Plant-Derived Metabolites on Prevention of Chronic Disease

Plant metabolites have been shown to have positive impacts on the prevention and amelioration of chronic diseases [10,13,21,24]. According to the World Health Organization, CVDs are the leading cause of death globally, 85% of which are due to heart attack and stroke. Risk factors for CVD include physical inactivity, tobacco use, harmful use of alcohol, and an unhealthy diet [69]. Plant-based metabolites and nutrients can be added to a healthy diet, and studies have shown mechanisms between some metabolites and the prevention of CVD and other chronic diseases [70].

Phenolic compounds are a class of metabolites found in various plant foods with antioxidant activity. Their ability to reduce oxidative stress within cells has indicated their possible function for the prevention of chronic diseases such as CVD. Upon evaluation of the ACE inhibitory activity and cytoprotective capacity, one study showed the potential of several phenolic acids as a component for the treatment of hypertension [71]. The phenolic acids showed significant protective activity for endothelial cells against oxidative damage by the mechanism of modulating nitric oxide, glutathione (GSH), malondialdehyde (MDA), and ROS levels. Specifically, the heme oxygenase-1 (HO-1), NADPH quinone oxidoreductase (NQO-1), glutamate-cysteine ligase catalytic subunit (GCLC), Phospho-Akt (p-Akt), and Phospho-eNOS (p-eNOS) expression levels were enhanced by sinapic acid, easing endothelial dysfunction [71]. Yu et al. [71] determined that the specific mechanism of the antihypertensive effect of sinapic acid was related to enhanced expression of antioxidant-related proteins and dose-dependently increased phosphorylation of endothelial nitric oxide synthase (eNOS) and activation of protein kinase B (Akt). Ultimately, the researchers determined that clinical studies will need to be conducted to assess the actual effects of phenolic acids in the human diet as treatment for CVD.

Diabetes is another chronic disease that is prevalent in the US population, with 11.6% of the population suffering from this condition. There are three types of diabetes with different causes. Type 1 is caused by an autoimmune reaction that causes the body to stop producing insulin, type 2 develops over several years when the body is not able to effectively utilize insulin to maintain blood sugar levels, and gestational diabetes develops in pregnant women who have never had diabetes [72]. There has been a significant amount of recent work regarding dietary strategies and utilization of plant-based bioactive compounds for the control and prevention of diabetes.

An in vitro study of hydroxy-isoflavone rich soybean extract presented evidence of inhibitory activity towards α-glucosidase and Dipeptidyl peptidase 4 (DPP4). Additionally, the modes of action of isoflavone extract were found to be significant inhibitors of sodium/glucose cotransporter 1 (SGLT1)-dependent glucose transport, causing a decrease in C-reactive protein (CRP) messenger RNA (mRNA) and secretion of protein levels in cultured Hep B3 hepatocytes. Thus, there is some evidence indicating antidiabetic mechanisms related to the intake of soy isoflavones [73].

Another study conducted on mice also demonstrated the modulation of an insulin-signaling pathway by soy isoflavone. The soy isoflavone biochanin A was given as a supplement to diabetic mice by feeding them a high-fat diet. For mice fed with the biochanin A, their body weight, plasma glucose, and insulin were significantly lower than control mice. The study determined that insulin receptor substrate 1, phosphoinositide 3-kinases (PI3K), protein kinase B (also known as Akt), and glucose transporter type 4 protein were all increased in skeletal muscle for mice supplemented with biochanin A. Overall, the study demonstrated the possibility of biochanin A as a treatment to improve insulin sensitivity through the activation of insulin signaling [74].

Diabetic kidney disease is a complication associated with chronic diabetes that is the most common cause of end-stage kidney disease globally. A study on the treatment of diabetic db/db mice with arctigenin and puerarin indicated a synergistic effect of these two isoflavones on diabetic kidney disease. The antioxidative stress and anti-inflammatory effects of these compounds may be related to their impact on the improvement or prevention of diabetic kidney disease. Arctigenin was found to inhibit nuclear factor-κB (NF-κB) phosphorylation, a primary proinflammatory pathway activated in human diabetic kidney diseased kidneys, through activation of Sirtuin 1 (Sirt1). A complementary anti-inflammatory effect was found for the treatment with puerarin and arctigenin, which are proposed to explain the additive renal protection seen in the mice with diabetic kidney disease [75]. Thus, the literature, along with the experimental evidence, has stated the evidence that plant bioactive compounds hold antioxidative, anti-inflammatory, and immunomodulatory attributes in addition to augmenting intestinal barrier functioning, thereby contributing to the mitigation of the pathological effect of several human disorders.

While some modes of action for plant metabolites have been investigated, many of the specific underlying mechanisms of their benefits remain unclear. Additional research is needed, especially in vivo and clinical studies, to better understand these mechanisms. The study of these mechanisms is complicated with complex interactions, especially in studies utilizing complex extracts or whole foods. While the study of specific compounds can be beneficial for understanding their role in human health and disease prevention, their mechanisms may be altered when consumed along with food or as part of whole foods in plant-based diets.

## 4. Why Are Plant-Based Compounds Associated with Low Risks for Chronic Diseases?

The preclinical and clinical outcomes specify that an appropriate diet has an advantageous effect on humans along with cardiopreventive attributes. Foods enriched with fruit and vegetables protect CVDs. Several plant-derived complexes, viz., fatty acids, peptides, polyphenols, oligosaccharides, and vitamins, also possess cardioprotective properties, thereby promoting heart condition [76] (Table 2; Figure 3).

### 4.1. Cardiovascular Health

Comprehensive scrutiny also validates the potential of plant extracts’ ability to regulate human blood pressure [19,106]. The experimental statistics advocating for grape polyphenols decrease the possibility of atherosclerosis via a mechanism, i.e., reduction in low-density lipoprotein oxidation [107]. Likewise, the polyphenols derived from olive oil, “hydroxytyrosol”, improve heart disorders by acting as a cardiopreventive agent owing to the strong radical scavenging attribute [95,108]. The mouse model study also demonstrated the cardioprotective effects of anthocyanin, where a mouse nourished with an anthocyanin-rich maize diet considerably reduced infarct mass, coronary occlusion, and reperfusion in comparison to the one fed an anthocyanin-free diet [109,110].

Considerations of prevention of CVD are especially important for aging populations due to chronic low-grade inflammation that develops upon advanced age and contributes to CVD risks. A study of participants on a diet supplemented with polyphenol-rich olive oil showed that arterial inflammation and atherosclerotic lesion microcalcification were reduced [111]. Olive oil is a rich source of phenolic compounds that have shown great evidence for promoting cardiovascular health and reducing the risk of CVD [64,111].

### 4.2. Gut Health

The gut microbiota population of an individual is a reflection of their intestinal health and plays a major role in overall health. Many plant metabolites, such as phenolic compounds, impact the gut health and microbiome. With long-term consumption of phenolic compounds from grape pomace, a shift in gut microbiota was found in a study on rats. Thus, preliminary evidence is provided that the modulation of gut bacteria phenotypes due to the supplementation with grape phenolic compounds has the potential to ameliorate adverse outcomes on the gut microbiome of aging persons [87]. Grapes are also a rich source of proanthocyanidins, which are a class of phenolic compounds that have been shown to have antibiotic activity. Proanthocyanidins from grape seeds were shown to have strong antioxidant activity and beneficial effects on gut health. A study on piglets showed that adding proanthocyanidins to their diet improved intestinal health via increased bacterial abundance and diversity and reduced the occurrence of diarrhea [89].

Tea is a common beverage for many people, and there is a plethora of evidence of its health-promoting effects. This extends to the effects of tea catechins on gut health as antioxidant, anticarcinogenic, antimicrobial, and antiviral compounds. A study utilizing epigallocatechin gallate (EGCG) and other key catechins found in oolong tea were found to impact in vitro fecal fermentation. The catechin extract promoted the growth of the *Lactobacillus–Enterococcus* group and *Bifidobacterium* spp. and increased total SCFA concentrations. The evidence presented gives a strong indication that these bioactive catechins from oolong tea have prebiotic potential and could contribute to gut health [96]. Overall, various plant foods are rich sources of metabolites with bioactive properties that promote gut health, which is an important factor in overall human health.

### 4.3. Anticancer

Epigenetic alterations may be produced by changing aspects of diet and lifestyle. Presently, identifying and developing epigenetically based interferences can be a critical anticancer approach [112]. A wide range of studies provide evidence that natural dietary compounds possess the potential for controlling pathways correlated with cancer stem cell pathways with self-renewal, for example, the Hedgehog, Notch, and Wnt/β-catenin pathways [113,114]. Plant-derived compounds like polyphenols also improve negative epigenetic mutations in the cancer cells, hindering carcinogenicity, stopping metastatic progression, and stimulating the tumor cells to chemotherapy or radiotherapy [115]. Several derivatives of plants, viz., anthocyanins, chalcones, flavanones, flavonols, and isoflavones, play a starring role as negative controllers of cell signaling pathways mediating proinflammatory responses, stimulating preventive measures of cancer [85,90,101,103].

Since eggplant is a great reservoir of phenolic components with strong antioxidative capability, perhaps it assists in detoxifying free radicals. Thus, eggplants could be employed in reducing the incidence of cancer. Additional research on eggplant also validates the application of alkaloid (solasodine) as a preventive approach toward lung cancer [116]. One study provided evidence that solasodine decreases the viability of the A549 cancer cell line and prevents cell invasion by inhibiting the expression of matrix metalloproteinase (MMP) matrix. Furthermore, it also reduced PI3K/Akt signaling, thereby downregulating microRNA-21 (miR-21) expression [85,117]. Additionally, other foods such as quinoa [98], sorghum [101], blueberry, and soybean [103] have been shown to contain bioactive components with anticancer potential. The evidence is clear that various plant metabolites act as anticancer agents, but further work is needed to determine their dietary function in the prevention of cancer and the possibilities of their supplementation as cancer treatment and prevention.

### 4.4. Antidiabetes

There has been extensive study on many plant foods and plant metabolites for their antidiabetic activity [80,92,105]. Blackcurrants are rich in anthocyanins that could be exploited to prevent diabetes. Anthocyanins from blackcurrants were shown to inhibit yeast α-glucosidase activity when dosed at lower concentrations than acabose, which is a common medication for the treatment of diabetes [80]. The study by Barik et al. [80] provided evidence of the potential of anthocyanins to affect postprandial hyperglycemia through their α-glucosidase inhibition and inhibitory function toward salivary α-amylase and glucose uptake and sugar transporters. The effects of the compounds from blackcurrants could have the potential to lower the risk of developing type 2 diabetes [80].

Red-skinned onion and purple eggplant are other sources of phenolic compounds with possible antidiabetic properties. Cattivelli et al. [84] found that cooking increased the bioavailability of phenolic compounds from the onion and eggplant, which improves their action against the risks of type 2 diabetes. The study found that certain compounds in the onion inhibited three of the key enzymes that are involved in diabetes pathogenesis [84]. Another study was conducted on the effects of novel peptides derived from quinoa. The bioactive peptides can be released during digestion and have compelling in vitro antidiabetic function [97]. Yang et al. [105] conducted a study on diabetic rats which showed improved glucose homeostasis as a result of compounds from fermented soybeans. Fasting serum glucose levels were at least partially restored with cooked soybeans and two types of fermented soy products. Additionally, the fermented soybean resulted in increased fasting insulin levels in the diabetic rats. Thus, in addition to beneficial plant metabolites in soybeans, fermentation may create additional bioactive compounds with antidiabetic properties [105]. The evidence of a variety of studies indicates that plants and plant-based diets are rich in highly beneficial antidiabetic compounds. The considerable task ahead is to identify and utilize these metabolites in a strategic manner to reduce the risks of diabetes and other chronic diseases.

## 5. Conclusions

The current literature review reinforces that plant-derived primary and secondary metabolites have attracted concern toward enhancing health owing to their diverse traits. The current appraisal elucidates plants’ imperative roles as reservoirs of numerous biologically active complexes providing multiple health-promoting benefits to humans. Moreover, a comprehensive analysis regarding biosynthesis, functioning, and the action mechanism of these biologically active complexes with respect to several stress circumstances is indispensable. Additionally, the different action mechanisms employed by these complexes concerning respective chronic disorders are imperative. This paper reviewed scientific corroboration associated with plant-derived bioactive constituents and chronic diseases, namely, blood pressure, brain health, diabetes, CVDs, cancer, gut health, lipid profile, and oxidative stress, concerning their regulation and inhibition. Plant complexes, viz., alkaloids, flavonoids, glycosides, polyphenols, polysaccharides, saponins, triterpenoids, etc., exhibited an anticarcinogenic asset. The compounds comprise repressive possessions on the proliferation of cancer cells, angiogenesis, and regulation of apoptosis. The primary function of these compounds is blocking the proliferation of tumor cells, further initiating apoptosis through the diverse passageway. The exploitation of such biologically active compounds from various plants is an effectual substitute for inhibiting cancer.

Until now, inadequate reports have been documented on human interference experiments that confirm auspicious traits of plant-derived metabolites. Thus, prior to their execution, additional meticulous clinical examinations are prerequisites. With the invention of novel molecular targets for corresponding diseases, this research area is instantly becoming noticed. Hence, a forthcoming investigation associated with human experimentation is essential to confirm their enhanced effect and suitability as curative agents. Moreover, authentication of this approach would provide chances for users to approach the profitable practice of plants in several ways, viz., medication, producing novel products, and implementing new dietary strategies.

## 6. Future Prospects

Plant-derived complexes hold a diversified pool of biologically active constituents, demonstrating distinctive biological traits. Owing to their nontoxic nature, these constituents are ideal for treating several diseases globally, but such a shift must be prudently perceived, as not entirely naturally occurring composites are advantageous. Likewise, several chronic disorder treatments have included the prevalent application of various vegetative medicines as harmonizing remedies. Consequently, each time, various novel cytotoxic constituents were obtained to institute different possible complexes for respective remedies. Consistent consumption of fruit, grains, and vegetables reduces the possibility of several metabolic patterns. Concerning this, extensive investigation into practicing conventional plants as a pool of biologically active complexes is indispensable.

Additional investigation and review of the safety of consumption of phytochemicals and nutrivigilance are especially important for their use as drugs or as nutritional supplements. It is of the utmost importance to understand the overall safety and safe dosage of each phytochemical [118]. Many consider phytochemicals from plants to be safe as they are derived from natural sources. However, toxicity cases have been reported as having varied potential for risk of disease. Factors impacting the safety of phytochemicals include extraction method, dosage, and drug interactions. There is very little investigation of the safety concerns regarding dietary or therapeutic use of many phytochemicals [119]. Future work is needed to assess safety concerns and implement appropriate regulation of phytochemicals.

These practices comprise obtaining plant-based composites in a purified formula for testing their efficacy against several diseases (in vitro and in vivo). Yet, certain concerns occur when investigating prominent authenticated plant-derived compounds for practice in several diseases. It is a multifaceted technique necessitating further cutting-edge analytic and scientific approaches aimed at recognizing prospective compounds, their bioavailability, and target tissue action. Still, various limitations remain to be addressed prior to product priming; punctilious investigation and clinical tests need to be executed, validating the dose competence that can postulate evidence on action mechanisms in humans internally. Finally, it might be noteworthy to understand and disclose how enriched eating routines contribute to a healthy life in addition to playing a remarkable part in inhibiting various illnesses.

## Figures and Tables

**Figure 1 foods-13-01020-f001:**
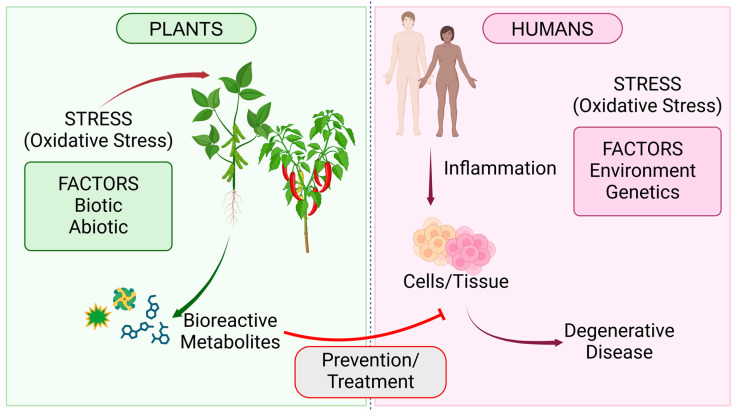
An inimitable association between plants and humans concerning stress (Created with BioRender.com, Accessed on 27 January 2024).

**Figure 2 foods-13-01020-f002:**
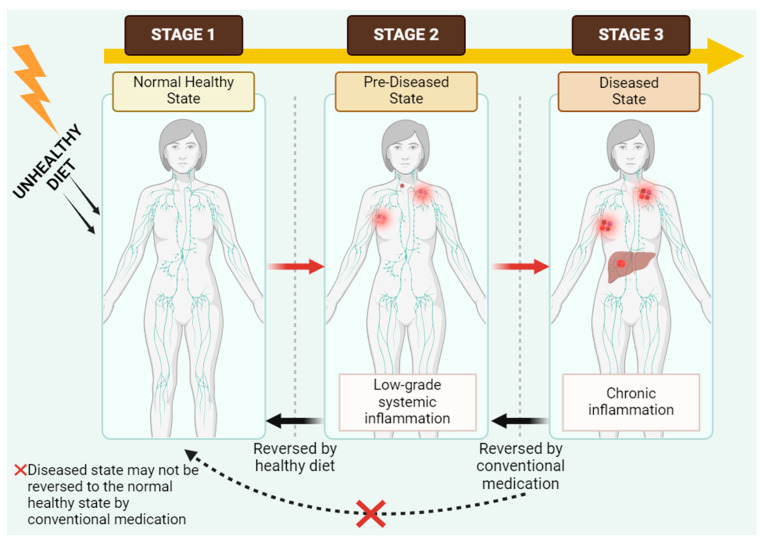
A holistic scheme portraying the association between the consumption of food and chronic diseases (Created with BioRender.com, Accessed on 27 January 2024). The red X indicates that the diseased state is not reversed to the normal healthy state by conventional medication.

**Figure 3 foods-13-01020-f003:**
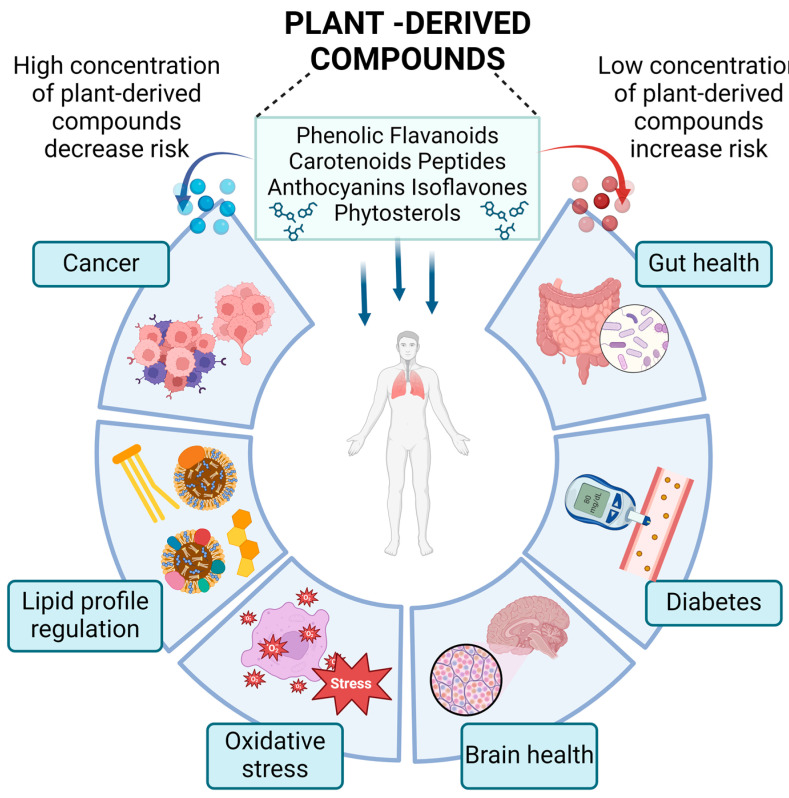
Multifarious role of plant-based compounds associated with the prevalence of chronic diseases (Created with BioRender.com, Accessed on 27 January 2024).

**Table 1 foods-13-01020-t001:** Principal dietary compounds concerning human health.

Types		Applications	
Phenolic compounds	Flavonoids	Antioxidant activity, anticancer activity, cognition	[10,11]
	Phenolic acids	Antioxidant activity, gut health, cardiovascular prevention	[12,13]
	Tannins	Antioxidant activity, anticancer activity	[14,15]
	Stilbenes	Anticancer activity	[15]
	Lignans	Antioxidant activity, Antidiabetic, gut health	[16,17,18]
	Phytoestrogens/ Isoflavones	Antioxidant activity, anticancer activity, reduction in LDL cholesterol levels	[19,20,21]
Carotenoids	β-carotene	Antioxidant activity, slowing macular aging	[22]
	Lycopene	Antioxidant activity, anticancer activity, reduction in cholesterol levels	[22]
	Lutein	Anticancer activity	[22]
Fatty acids	Omega 3 fatty acids	Anti-inflammatory activity, reduction in cholesterol disposition, maintain brain functioning	[23,24]
Proteins	Bioactive peptides	Metabolism, antioxidant activity, antihypertensive	[25]
Fat-soluble	Vitamin A	Antioxidant activity, prevent several cancers, maintain skin, and eye mucous membrane healthy	[26]
	Vitamin D	Regulate bone and teeth formation, helps in absorption of calcium	[26]
	Vitamin E	Antioxidant activity, enhances immune system, helps in formation of nerve tissue	[26]
	Vitamin K	Essential for blood clotting	[26]
Water-soluble	Vitamin C	Immune modulating effects	[27,28]
	Vitamin B1, B2, B3, B6, B12	Neurological functioning, energy conversion, helps in production of cells genetic material, red blood cell formation, regulate metabolism	[29,30]
	Foliate	Prevent birth defects, red blood cell formation, cardio-preventive activity	[31]
Minerals	Calcium	Essential for bones and teeth formation, glandular functioning	[32]
	Iron	Energy production, transportation of oxygen, maintenance of red blood cells and hemoglobin, prevention of iron-deficiency anemia	[32,33]
	Magnesium	Essential for bone formation, muscle functioning, prevent premenstrual syndrome	[32]
	Phosphorus	Essential for genetic material, healthy bones, and teeth, produce and store energy	[32]
Trace elements	Cobalt	Component of Vit B12, B12 coenzymes	[32]
	Chromium	Conversion of carbohydrates and fats into energy	[34]
	Copper	Essential for production of hemoglobin and collagen, heart functioning	[32]
	Iodine	Thyroid proper functioning	[32]
	Selenium	Heart function, antioxidant activity	[32]
	Zinc	Cell reproduction, immunity, appetite	[32]
Other bioactive	Biotin	Required for metabolic functioning	[32]
	Choline	Synthesis of phospholipid membrane, cell function	[35]

**Table 2 foods-13-01020-t002:** Promising health benefits associated with bioactive compounds derived from plants.

Plant	Bioactive Compounds	Health Benefits	
**Adzuki beans (*Vigna angularis*)**	Polyphenols	Gut health	[77]
**Almonds (*Prunus amygdalus*)**	Fatty acids, polyphenols	Metabolic recovery	[78]
**Banana (*Musa acuminata*)**	Serotonin	Mood stabilizing	[79]
**Black currants (*Ribes nigrum*)**	Anthocyanins	Antidiabetic activity	[80]
**Cranberry (*Vaccinium macrocarpon*)**	Polyphenols	Gut health	[81,82]
**Cruciferous vegetables**	Glucosinolates	Anti-inflammatory activity	[83]
**Eggplant (*Solanum melongena*)**	Phenolic compounds	Antidiabetic activity	[84]
	Solasodine	Anticancer activity	[85]
**Ginger (*Zingiber officinale*)** **and mulberry fruit (*Morus alba*)**	Extract	Brain health and oxidative stress	[86]
**Grape (*Vitis* spp.)**	Phenols, proanthocyanididns	Gut health	[87,88,89]
**Hemp grain (*Cannabis sativa*)**	Phenolics	Antioxidant activity	[90,91]
**Lettuce (*Lactuca sativa*)**	Polyphenols	Antidiabetic activity	[92]
**Manchurian walnut** **(*Juglans mandshurica*)**	Hydrolyzed peptides	Antidiabetic activity	[93]
**Oats (*Avena sativa*)**	Phenolics	Antioxidant, prebiotic	[12,94]
**Olive oil (*Olea europaea*)**	Polyphenols	Cardiovascular activity	[95]
**Oolong tea (*Camellia sinensis*)**	(−)-epigallocatechin	Gut health	[96]
**Quinoa (*Chenopodium quinoa*)**	Bioactive peptides	Antidiabetic and anticancer	[97,98]
**Seaweed (*Porphyra tenera*)**	Extract	Gut health, antioxidant	[99,100]
**Sorghum (*Sorghum bicolor*)**	Phenolic	Anticancer activity	[101,102]
**Soybean (*Glycine max*)**	Isoflavones	Anticancer activity	[103,104]
	Isoflavonoid	Insulinotropic activity	[105]

## Data Availability

No new data were created or analyzed in this study. Data sharing is not applicable to this article.

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
