# Peer review of "Examination of Primary and Secondary Metabolites Associated with a Plant-Based Diet and Their Impact on Human Health"

_foods, 2024, doi:10.3390/foods13071020_

Round 1

Reviewer 1 Report

Comments and Suggestions for Authors

Dear Authors, your article is very interesting, and your images are complete, however, you need to deepen the discussion. Phenolics possess notable anti-inflammatory, antioxidant and antidiabetic properties, and so on. You need to innovate. The health gut and all the topics need to be more exploited and, please, cite the newest studies.

Line 13: “has become a burgeoning trend. Foods eaten as part”.

And biotic factors do not change secondary metabolites?

Table 1: Isoflavones, including phytoestrogens, are also phenolic compounds?

Lines 68/ 207: Polyphenols include flavonoids and anthocyanins, and carotenoids are not polyphenols – rewrite this sentence, please.

Trace elements that are minerals, no?

Line 138: plants only synthesize certain chemical complexes aiding in protection and adaptation against abiotic stressors? Clarify this sentence, please.

Vitamin B is also useful in neurological pathologies, like Parkinson’s disease.

Lines 389 and 390: “According to the World Health Organization, CVDs”

Lines 383 to 385: “Consuming plant foods rich in phytochemicals, such as phenolic compounds,”

Line 387: isoflavones are phenolic compounds.

The declaration of interest is not complete.

And toxicity? What are the advantages of using phytochemicals and nutrients instead of chemical drugs? Explore all the topics to richer the article.

Comments on the Quality of English Language

The English needs to be revised.

Author Response

Response to Reviewers

Reviewer 1

Dear Authors, your article is very interesting, and your images are complete, however, you need to deepen the discussion. Phenolics possess notable anti-inflammatory, antioxidant and antidiabetic properties, and so on. You need to innovate. The health gut and all the topics need to be more exploited and, please, cite the newest studies.

Response: Thank you for the review of our manuscript. We have addressed your specific comments below. We have made revisions to the manuscript to innovate and exploit the topics of gut health and others. We have cited a number of the newest studies. Our revision has included additional studies published very recently during 2024.

Line 13: “has become a burgeoning trend. Foods eaten as part”.

Response: This has been revised to “Consumption of plant-based diets have become a burgeoning trend, and are increasingly consumed globally owing to their substantial energy intensity and dietetic advantages.”

And biotic factors do not change secondary metabolites?

Response: We understand that both abiotic and biotic factors impact secondary metabolites. We have revised this sentence to make this more clear. “In a nutshell, plant-derived metabolites perform indispensable roles in the plant defense system against abiotic and biotic environmental stresses. Abiotic stress includes temperature, light, wounding, chemical derivatives, and so on, whereas biotic stress is associated with microbes, insects, and animals [6].”

Table 1: Isoflavones, including phytoestrogens, are also phenolic compounds?

Response: This category has been moved under the phenolic compounds section. (See table 1)

Lines 68/ 207: Polyphenols include flavonoids and anthocyanins, and carotenoids are not polyphenols – rewrite this sentence, please.

Response: Line 207 refers to compounds which are all phytochemicals or secondary metabolites not compounds which are grouped under polyphenols. We have re-written the sentence on line 68 as follows.

“Plant-derived antioxidants mainly polyphenols, for example, flavonoids, lignans, stilbenes, and phenolic acids, as well as carotenoids, anthocyanins, vitamins C, and vitamin E, possess a comprehensive array of biological traits, anti-aging, anti-cancer, anti-infectious, anti-inflammatory, anti-viral, etc. [34,35].”

Trace elements that are minerals, no?

Response: While trace elements are minerals, they have their own category as the human body needs trace minerals in significantly lower amounts. The requirements for bodily functions of trace elements are even lower than that of the small amount of minerals needed. For example humans need 1000-1300 mg of calcium per day but only require 90-220 ug of iodine per day. Additionally, high dietary intake of some trace elements can have negative health consequences. For example high intake of copper in a diet can cause liver damage, abdominal pain, cramps, nausea, diarrhea, and vomiting.

Line 138: plants only synthesize certain chemical complexes aiding in protection and adaptation against abiotic stressors? Clarify this sentence, please.

Response: We have revised this sentence and added additional information for clarity.

“Principally, when plants experience stress (e.g., oxidative stress), they synthesize certain chemical complexes which aid in protection and adaptation against those stressors. For example, plants under salt stress can produce metabolites, such as proline, hydroxyproline, glycine betaine, sugars, or polyamines to maintain low intracellular osmotic potential [44]. Another example is the production of phenolic compounds in plant tissue as part of an adaptive response to stressors. Stress conditions which may illicit production of phenolic compounds in plants include low temperature, pathogen infection, nutrient deficiency, and herbivores [45].”

Vitamin B is also useful in neurological pathologies, like Parkinson’s disease.

Response: We have included additional information regarding B vitamins and Parkinson’s disease

“There is also evidence of a connection between B vitamins and Parkinson’s disease. Parkinson’s disease symptoms may worsen for those having low B12 levels [57]. Another study showed that while long term elevated intake of vitamin B6 or folate does not reduce risk of Parkinson’s disease, there may be a possible protective effect from vitamin B12 [58].”

Lines 389 and 390: “According to the World Health Organization, CVDs”

Response: We have revised this statement as suggested.

Lines 383 to 385: “Consuming plant foods rich in phytochemicals, such as phenolic compounds,”

Response: We have revised this statement as suggested.

Line 387: isoflavones are phenolic compounds.

Response: We have removed the specific reference to isoflavones here, as suggested.

The declaration of interest is not complete.

Response: We have completed the declaration of interest as such: “Declaration of interest: The authors declare that they have no known competing financial interests or personal relationships that could have appeared to influence the work reported in this article.”

And toxicity? What are the advantages of using phytochemicals and nutrients instead of chemical drugs? Explore all the topics to richer the article.

Response: Thank you for this excellent suggestion. The majority of the suggested topic is outside the current scope of this review paper, thus we chose to highlight the key points and discuss the need for additional work in this area. We have included some discussion of this topic in the future prospects section ”Additional investigation and review of safety of consumption of phytochemicals and nutrivigilance. This is especially true for their use as drugs or as nutritional supplements. It is of the utmost importance to understand the overall safety and safe dosage of each phytochemical [118]. Many consider phytochemicals from plants to be safe as they are derived from natural sources. However, toxicity cases have been reported having varied potential for risk of disease. Factors impacting safety of phytochemicals include extraction method, dosage, and drug interactions. There is very little investigation of the safety concerns regarding dietary or therapeutic use of many phytochemicals [119]. Future work is needed to assess safety concerns and implement appropriate regulation of phytochemicals.”

Reviewer 2 Report

Comments and Suggestions for Authors

The selected topic is very wide and encompasses vital facets of plant-based diets and their health ramifications. The article is structured well, however, needs certain revision to make article more informative:

1. Author representing the broad-spectrum effect of plant-based metabolites and phytoconstituents. It would be better if the author could focus on some important plant-based metabolites and elucidate their health effects in detail.

2.  Additionally, it is crucial to offer a comprehensive explanation (detailed mode of action) of the mechanisms by which plant-based metabolites contribute to various health-promoting activities, including cardiovascular health, diabetes management, gut health, and anticancer properties.

Comments on the Quality of English Language

Minor english editing

Author Response

Reviewer 2

The selected topic is very wide and encompasses vital facets of plant-based diets and their health ramifications. The article is structured well, however, needs certain revision to make article more informative:

Response: Thank you for your review of our manuscript. We have conducted revisions to make the article more informative. We have addressed your comments below.

Author representing the broad-spectrum effect of plant-based metabolites and phytoconstituents. It would be better if the author could focus on some important plant-based metabolites and elucidate their health effects in detail.

Response: Thank you for the insightful comments. Our aim was to provide a holistic overview of plant metabolites and phytonutrients as part of plant based diets and their impact on human health. We have done our best to focus on a few key areas regarding this topic.

Additionally, it is crucial to offer a comprehensive explanation (detailed mode of action) of the mechanisms by which plant-based metabolites contribute to various health-promoting activities, including cardiovascular health, diabetes management, gut health, and anticancer properties.

Response: We agree that the mechanisms and detailed mode of action are important to understand the health promoting activities of phytochemicals. However, the majority of this topic is outside the scope of the current review paper. Our aim was to provide a holistic overview of the contribution of plant based diets to dietary consumption of their metabolites and phytochemicals.

Reviewer 3 Report

Comments and Suggestions for Authors

The manuscript is interesting and logically constructed. I have a few, minor comments:

line 339: do not list all authors, just provide: details of the first author et al.; check it throughout the text - similar situations are also in other places (e.g. lines 537 and 543) - change this

lines 421-424: there are abbreviations used here that were not previously explained; check the entire text to see if there are similar situations elsewhere

lines 454: (Figure 3; Table 2): change the order to (Table 2; Figure) - this will match the order in which table 2 and figure 3 are placed in the manuscript

Author Response

Reviewer 3

The manuscript is interesting and logically constructed. I have a few, minor comments:

Response: Thank you for the review of our manuscript. We have addressed your comments below.

line 339: do not list all authors, just provide: details of the first author et al.; check it throughout the text - similar situations are also in other places (e.g. lines 537 and 543) - change this

Response: We have corrected the in text citation format according to your comment and the journal guidelines.

lines 421-424: there are abbreviations used here that were not previously explained; check the entire text to see if there are similar situations elsewhere

Response: We have added the definitions to these abbreviations in the text. Akt is not an abbreviation, however it is also know as Protein kinase B, which we have indicated in the text. We have checked the text and included definitions for all abbreviations.

lines 454: (Figure 3; Table 2): change the order to (Table 2; Figure) - this will match the order in which table 2 and figure 3 are placed in the manuscript

Response: We have revised this as suggested.

Reviewer 4 Report

Comments and Suggestions for Authors

In the present manuscript, the authors present the potential of some plant compounds in the prevention and treatment of different chronic illnesses. Please see my comments below:

1.      Please update the applications from Table 1. For example:

·        Vitamin C is an antioxidant and has immune-modulating effects

https://www.mdpi.com/2072-6643/9/11/1211

-            Iron is involved in the synthesis and maintenance of optimal levels of red blood cells and haemoglobin and preventing iron-deficiency anaemia

https://www.ncbi.nlm.nih.gov/pmc/articles/PMC3999603/

2.      Lines 62-65: the classification of plants is not correct. Farm-produced vegetables are included in the legumes category … Please revise this paragraph.

3.      Phytochemicals could have some disadvantages. Authors must present issues regarding the safety of the phytochemicals and nutrivigilance activity

4.      Lines 160-161: Some examples of plant sources of anti-nutritional factors should be given by the authors.

5.      Lines 174-175: The authors should specify what type of risk they are referring to when they state "risk of type 2 diabetes”. Is it about the risk of onset or exacerbation?

6.      Table 2 - please add the scientific name (binomial name) of plants

Author Response

Reviewer 4

In the present manuscript, the authors present the potential of some plant compounds in the prevention and treatment of different chronic illnesses. Please see my comments below:

Response: Thank you for the review of our manuscript. We have addressed your specific comments below.

Please update the applications from Table 1. For example:

Vitamin C is an antioxidant and has immune-modulating effects https://www.mdpi.com/2072-6643/9/11/1211

Iron is involved in the synthesis and maintenance of optimal levels of red blood cells and haemoglobin and preventing iron-deficiency anaemia https://www.ncbi.nlm.nih.gov/pmc/articles/PMC3999603/

Response: Thank you for these suggestions. We have updated the table with these references and other relevant citations.

Lines 62-65: the classification of plants is not correct. Farm-produced vegetables are included in the legumes category … Please revise this paragraph.

Response: We have revised this sentence for clarity. Legumes are defined as plants are plants that produce their fruit as pods. We refer to vegetables as other edible plant material such as roots, stems, or leaves.

Phytochemicals could have some disadvantages. Authors must present issues regarding the safety of the phytochemicals and nutrivigilance activity

Response: Thank you for your insightful comment. We have focused the scope of our review on The scope of this review includes (i) beneficial role of biochemical active composites in plants and humans, (ii) plant-based compounds association with the incidence of chronic diseases, (iii) action mechanism accountable for the consumption of plant-derived compounds with major health outcomes, (iv) exploration of why and how plant-based compounds associated with a low prevalence of chronic disorders: prospective and pragmatic associations.

However, we find your suggestion pertinent to future research and review and have included this information in section 6 “Additional investigation and review of safety of consumption of phytochemicals and nutrivigilance. This is especially true for their use as drugs or as nutritional supplements. It is of the utmost importance to understand the overall safety and safe dosage of each phytochemical [118]. Many consider phytochemicals from plants to be safe as they are derived from natural sources. However, toxicity cases have been reported having varied potential for risk of disease. Factors impacting safety of phytochemicals include extraction method, dosage, and drug interactions. There is very little investigation of the safety concerns regarding dietary or therapeutic use of many phytochemicals [119]. Future work is needed to assess safety concerns and implement appropriate regulation of phytochemicals.”

Lines 160-161: Some examples of plant sources of anti-nutritional factors should be given by the authors.

Response: We have included some examples of plant sources of anti-nutritional factors.

Lines 174-175: The authors should specify what type of risk they are referring to when they state "risk of type 2 diabetes”. Is it about the risk of onset or exacerbation?

Response: We have revised this statement to state ” There is some evidence that the risk of exacerbating type 2 diabetes may be reduced by intake of plant-based proteins.”

Table 2 - please add the scientific name (binomial name) of plants

Response: We have added the binomial name of the plants in table 2.

Reviewer 5 Report

Comments and Suggestions for Authors

Thanks for the opportunity to review this manuscript, entitled „Examination of primary and secondary metabolites associated with a plant-based diet and their impact on human health”. The subject of the manuscript is topical, but I recommend the necessary corrections.

The text provides a comprehensive overview of the role of plant-derived compounds in human health and disease prevention. It effectively explores the biochemical complexity of plant compounds and their potential impact on chronic disorders. The review identifies a gap in knowledge by highlighting the fragmented information available regarding the biochemical complexes of plants and their effects on human health at biological and molecular levels.

While the text does not explicitly mention similar reviews published recently, it addresses an important and evolving topic that continues to be relevant and of interest to the scientific community. The information presented sheds light on the association between plant compounds and human health in a detailed manner.

The cited references predominantly include recent publications within the last five years, which enhances the credibility of the review. There is no indication of any relevant citations being omitted, and the text maintains a balance without an excessive number of self-citations.

The statements and conclusions drawn in the text are coherent and well-supported by the listed citations. The review effectively links the role of plant-based compounds with chronic disorders and human health outcomes.

The tables and figures are well done and succinctly describe the necessary information. The appropriate visuals are easy to interpret, and it further strengthen the impact of the review.

Suggestions for Improvement:

Ensure that additional relevant citations are included to further substantiate the arguments.

Clearly outline the objectives and scope of the review at the beginning for better clarity.

Consider providing a summary or conclusion section to reinforce key findings.

Overall, the text provides valuable insights into the relationship between plant compounds and human health, making it a relevant and informative piece for researchers in the field.

Author Response

Reviewer 5

Thanks for the opportunity to review this manuscript, entitled „Examination of primary and secondary metabolites associated with a plant-based diet and their impact on human health”. The subject of the manuscript is topical, but I recommend the necessary corrections.

The text provides a comprehensive overview of the role of plant-derived compounds in human health and disease prevention. It effectively explores the biochemical complexity of plant compounds and their potential impact on chronic disorders. The review identifies a gap in knowledge by highlighting the fragmented information available regarding the biochemical complexes of plants and their effects on human health at biological and molecular levels.

While the text does not explicitly mention similar reviews published recently, it addresses an important and evolving topic that continues to be relevant and of interest to the scientific community. The information presented sheds light on the association between plant compounds and human health in a detailed manner.

The cited references predominantly include recent publications within the last five years, which enhances the credibility of the review. There is no indication of any relevant citations being omitted, and the text maintains a balance without an excessive number of self-citations.

The statements and conclusions drawn in the text are coherent and well-supported by the listed citations. The review effectively links the role of plant-based compounds with chronic disorders and human health outcomes.

The tables and figures are well done and succinctly describe the necessary information. The appropriate visuals are easy to interpret, and it further strengthen the impact of the review.

Response: We thank you for the review of this manuscript. We have responded to your specific comments below.

Suggestions for Improvement:

Ensure that additional relevant citations are included to further substantiate the arguments.

Response: We have included additional relevant citations of recent literature throughout the manuscript to enhance the arguments of this review paper.

Clearly outline the objectives and scope of the review at the beginning for better clarity.

Response: We have added this information to the introduction section. “Therefore, in this review paper, our objective is to synopsize the portrayal of the association of several plant compounds with protection in relation to the cause and the preclusion of several chronic disorders.” “The scope of this review includes (i) beneficial role of biochemical active composites in plants and humans, (ii) plant-based compounds association with the incidence of chronic diseases, (iii) action mechanism accountable for the consumption of plant-derived compounds with major health outcomes, (iv) exploration of why and how plant-based compounds associated with a low prevalence of chronic disorders: prospective and pragmatic associations.”

Consider providing a summary or conclusion section to reinforce key findings.

Response: We have included a conclusion section (see section 5.) to provide a summary and reinforce the key findings of this review.

Overall, the text provides valuable insights into the relationship between plant compounds and human health, making it a relevant and informative piece for researchers in the field.

Response: Again, thank you for your time in reviewing our manuscript. We have made the revisions based on all of the reviewer comments to improve the value of this paper.

Round 2

Reviewer 1 Report

Comments and Suggestions for Authors

thank you for your reply, nice work!

Author Response

Thank you for the review of our manuscript.

Reviewer 2 Report

Comments and Suggestions for Authors

The article has been revised as per the raised concer, I endorse the publication of the article

Comments on the Quality of English Language

NA

Author Response

Thank you for your review of this manuscript. We have conducted a final review of the English language grammar.